# Impact of *Plasmodiophora brassicae* on Canola Root and Rhizosphere Microbiomes and Its Implications for Clubroot Biocontrol

**DOI:** 10.3390/pathogens14090904

**Published:** 2025-09-09

**Authors:** Jorge Cordero-Elvia, Leonardo Galindo-González, Rudolph Fredua-Agyeman, Sheau-Fang Hwang, Stephen E. Strelkov

**Affiliations:** 1Department of Agricultural, Food and Nutritional Sciences, University of Alberta, Edmonton, AB T6G2P5, Canada; jcordero@ualberta.ca (J.C.-E.); leonardo.galindogonzalez@inspection.gc.ca (L.G.-G.); freduaag@ualberta.ca (R.F.-A.); sh20@ualberta.ca (S.-F.H.); 2Ottawa Plant Laboratory, Science Branch, Canadian Food Inspection Agency, 3851 Fallowfield Road, Ottawa, ON K2H8P9, Canada

**Keywords:** biological control, clubroot, microbiome, resistance, rhizosphere, root, susceptibility

## Abstract

Clubroot, caused by the obligate parasite *Plasmodiophora brassicae*, is a soilborne disease affecting canola (*Brassica napus*) and other crucifers. Although planting resistant cultivars remains the primary strategy for managing clubroot, the emergence of resistance-breaking *P. brassicae* pathotypes continues to threaten canola production. In this context, soil and root microorganisms may play a role in suppressing the disease. This study investigated the impact of *P. brassicae* infection on the microbial communities of soil, seeds, roots, and the rhizosphere in susceptible and resistant canola lines, with the aim of analyzing host–pathogen–microbiome interactions and identifying microbial taxa potentially associated with disease resistance. Our findings showed that resistant canola lines inoculated with *P. brassicae* (pathotype 3A) exhibited reduced disease severity compared to their susceptible counterparts. Diversity analyses of microbial communities revealed that clubroot-resistant canola lines tended to maintain more stable and diverse fungal communities, with a higher Shannon index than susceptible lines. Inoculation with *P. brassicae* induced more pronounced changes in the root microbiome than in the rhizosphere. Additionally, the seed microbiomes of resistant and susceptible lines displayed distinct bacterial and fungal profiles, suggesting that clubroot susceptibility may influence seed-associated microbial community composition. Differential abundance analysis of root and rhizosphere microbiomes indicated that certain microbial taxa, including bacterial genera such as *Acidovorax*, *Bacillus*, *Cupriavidus*, *Cytophaga*, *Duganella*, *Flavobacterium*, *Fluviicola*, *Luteimonas*, *Methylotenera*, *Pedobacter*, and *Peredibacter*, as well as fungal genera such as *Aspergillus*, *Candida*, *Fusicolla*, *Paecilomyces*, and *Rhizophlyctis*, may be recruited or enriched in resistant canola lines following *P. brassicae* inoculation, potentially contributing to reduced clubroot severity.

## 1. Introduction

Clubroot is a disease caused by the soilborne parasite *Plasmodiophora brassicae* Woronin (Plasmodiophorida: Plasmodiophoridae), which primarily affects members of the Brassicaceae family, including canola (*Brassica napus* L.) [1]. The most characteristic symptom of the disease is the formation of large, swollen galls or clubs on the roots, which severely impair root system function. Infection can lead to significant economic losses due to reduced crop yields and increased management costs. Practices such as crop rotation, sanitization of field equipment, and the application of fungicides and soil amendments have been recommended for clubroot management [2]. However, many of these approaches are difficult to implement or costly in canola production systems [3], making resistant cultivars the most effective method for control [4,5,6].

Recently, highly virulent pathotypes of *P. brassicae* capable of overcoming clubroot resistance in canola have emerged, with pathotype 3A being one of the most prevalent in western Canada [7,8]. The emergence of these resistant-breaking pathotypes poses a significant challenge to canola production. Therefore, identifying new control strategies may help improve disease management. In this context, the interplay between plant resistance mechanisms and associated microbial communities remains an area of active research. Several studies have highlighted the impact of plant-pathogen interactions on the structure and function of rhizosphere microbial communities. The rhizosphere and endophytic microbiomes host a diverse array of beneficial microorganisms that enhance plant defense by priming immune responses, producing antibiotics, and competing with or inhibiting plant pathogens [9,10,11]. These microbial communities can also improve soil quality, modulate nutrient availability, and promote plant growth [9,10].

Previous studies have shown that individual strains of bacterial and fungal species exhibit antagonistic effects against *P. brassicae*, including species of *Bacillus*, *Gliocladium*, *Lysobacter*, *Streptomyces*, *Trichoderma*, and *Zhihengliuella* [12,13,14,15,16,17]. For example, the application of a biofungicide containing *Bacillus subtilis* completely suppressed *P. brassicae* infection in canola through antibiosis and the induction of host resistance [12]. Similarly, the fungal species *Trichoderma guizhouense* and *Trichoderma koningiopsis* reduced clubroot severity by 44.3% and 57.3%, respectively [16]. A study investigating the impact of the soil microbiome on *P. brassicae* during pathogenesis found that soils with higher bacterial and fungal diversity were associated with reduced clubroot severity in *B. napus*, compared to soils with low microbial diversity [18]. Likewise, another study revealed that microbial diversity and relative abundance in clubroot-asymptomatic *B. napus* roots were greater than in symptomatic roots [19]. These findings suggest that pathogen suppression is influenced by entire microbial communities rather than individual species, emphasising the importance of microbiome for disease control.

Our previous research revealed that bacterial and fungal microbiomes associated with the rhizosphere and root endosphere of clubroot-susceptible (CS) and clubroot-resistant (CR) canola lines exhibited changes in microbial community structure and taxonomic composition in response to inoculation of two *P. brassicae* isolates, classified as pathotype 3A, with differential virulence [20]. Pathotype 3A is highly virulent and capable of overcoming resistance in many canola cultivars grown in Canada [7,8]. Our findings showed that disease severity increased when a *P. brassicae* isolate, previously cycled multiple times through clubroot-resistant (CR) canola lines, was used to inoculate a CR line [20]. This reduction in resistance is likely driven by the selection and amplification of more virulent pathogen genotypes [21,22,23]. The resulting changes in resistance levels also altered the microbial composition of the rhizosphere and root microbiomes, highlighting the need to complement resistance breeding with additional management strategies to sustain the effectiveness of clubroot resistance in canola. Based on these findings, we hypothesized that microbial taxa showing higher relative abundance in response to *P. brassicae* inoculation in CR lines may play a potential role in clubroot suppression.

In the current study, we assessed the effects of *P. brassicae* on the microbial structure and diversity of the soil, seeds, roots, and rhizosphere associated with three CR and three CS canola lines. We also examined the differential responses to clubroot of microbial communities among canola lines to identify bacterial and fungal taxa that may contribute to disease suppression. Metagenomic analyses indicated that *P. brassicae* inoculation induced more significant changes in the root microbiome compared to the rhizosphere fungal communities. Additionally, fungal diversity was higher in CR lines than in CS lines. Differential abundance analysis of the root and rhizosphere microbiomes identified bacterial and fungal taxa that were more abundant in CR lines after *P. brassicae* inoculation, indicating their potential role in clubroot biocontrol. These findings offer deeper insights into the complex interactions between plant resistance, pathogen dynamics, and microbial communities. Ultimately, this work contributes to the growing understanding of plant-microbe-pathogen interactions and supports the development of microbiome-based strategies for clubroot management.

## 2. Materials and Methods

### 2.1. Plant, Soil and Pathogen Material

We used six double-haploid *B. napus* lines that were either susceptible or resistant to *P. brassicae* pathotype 3A. The three resistant canola lines were 12PH0254, 12PH0271, and 12PH0277, while the three susceptible lines were 12PH0244, 12PH0245, and 12PH0247. All lines were derived from F1 plants of the cross between 11SR0099 (clubroot-resistant) and 12DH0001 (clubroot-susceptible). The seeds were not treated with fungicides or with any chemical or biological agents. For clarity throughout the manuscript, the resistant lines were designated as TR254, TR271, and TR277, and the susceptible lines as TS244, TS245, and TS247. The canola cultivar ‘Westar’ was included in the clubroot disease assessments as a susceptible control to confirm the viability of the pathogen inoculum and the suitability of conditions for disease development. Soil for the experiment was collected on August 6, 2021 from an organic field at the South Campus of the University of Alberta, which had no prior history of clubroot. The soil was sampled at a depth of 15 cm, was air-dried, and sieved to 4 mm. It was then mixed with Sunshine LA4 potting mixture (Sunshine Growers, Vancouver, BC, Canada) in a 2:1 (*v*:*v*) ratio of Sunshine mix to field soil and homogenized in a soil mixer for 1 h.

The *P. brassicae* isolate used for inoculation was selected from a field population collected from a commercial canola crop in Sturgeon County, Alberta, Canada. This isolate was classified as pathotype 3A using the Canadian Clubroot Differential (CCD) set [8]. The inoculum was prepared by macerating root galls in a variable-speed blender at high speed for 3 min, followed by two additional 2-min blending steps. The resulting spore suspension was filtered through three layers of cheesecloth, quantified using a hemocytometer, and adjusted to a final concentration of 1 × 10^7^ resting spores mL^−1^.

### 2.2. Experimental Design and Greenhouse Conditions

Canola seeds were germinated in Petri dishes on filter paper moistened with 1.5 mL of autoclaved distilled water for one week prior to inoculation. Seedlings were inoculated by dipping their root systems into the *P. brassicae* resting spore suspension for 15 s. A single inoculated seedling was transferred to a Cone-tainer (21 cm long, 3.8 cm in diameter; Stuewe and Sons, Inc., Tangent, OR, USA) filled with 100 g of the soil mixture. To maintain high disease pressure, an additional 1 mL of spore inoculum was applied to the base of each plant. The Cone-tainers were then placed in a greenhouse maintained at 22 °C with a 16 h photoperiod, using natural light supplemented with artificial lighting. The soil mix was kept saturated with water (pH 6.5) for the first week after inoculation, and plants were subsequently watered and fertilized (20 N: 20 P: 20 K) as needed. Control treatments did not receive inoculum.

Samples were collected at 7, 21, and 35 days after inoculation (dai) to assess microbial diversity and abundance. Three biological replicates, consisting of 24 plants each, were included for each harvest time, canola line, and treatment (inoculated and non-inoculated). Additionally, three more replicates per treatment were maintained for a final evaluation of clubroot disease severity at 42 days after inoculation (dai), as described below. Biological replicates were arranged in a completely randomized design and rotated on each harvest day, as shown in Appendix A.

The experiment was conducted in three rounds, under similar conditions, with each round including a different combination of CR and CS canola lines. The first round, comprising the canola lines TS244 and TR277, was conducted between August and September 2022. The second round, which included TS245 and TR254, was carried out between November 2022 and January 2023. The third round, involving TS247 and TR271, was conducted between February and April 2023.

### 2.3. Sample Collection

At each harvesting time, 72 Cone-tainers were randomly selected for each canola line and treatment. Plants with attached soil were extracted from the pots using a spatula. Shoots were separated from the roots, and loosely attached soil was removed from the roots by gentle shaking. Roots with adhering rhizosphere soil from plants within the same replicate were pooled together in a labelled plastic bag. Samples were stored at 4 °C until processing.

Rhizosphere soil was separated from roots by transferring the roots to a flask containing a sterile diluent (NaCl, KH2PO4, Na2HPO4, MgSO4, gelatin) at a ratio of root weight (g) to solution volume (mL) of 1:100 [24,25]. Roots were shaken on a rotary shaker at 200 rpm for 20 min [24]. The resulting soil/phosphate-buffered saline (PBS) solution was centrifuged for 10 min at 8000× *g* to collect rhizosphere soil, and the supernatant was decanted [22]. Roots were then washed with sterile water and surface-disinfested by immersion in a NaClO (1.05% *v*/*v*) prepared in sterile diluent at a root weight to solution volume ratio of 1:40 [23]. The roots were agitated on a rotary shaker (200 rpm) for 10 min, followed by four rinses with sterile diluent to remove the disinfectant. The efficacy of surface sterilization was verified by spreading 0.1 mL of the final wash onto Petri dishes containing 1/10-strength tryptone soy agar medium [26]. Roots were then frozen in liquid nitrogen and stored at −80 °C for molecular analyses.

To assess the initial microbial composition of the soil mixture and canola seeds, 5 g of soil samples were collected in buffer using the same procedure as for rhizosphere soil extraction. Surface-sterilized seeds (1 g) of each canola line were hydrated in sterile water for 4 h prior to disruption for DNA extraction. Three replicates were collected for both the soil mixture and the seeds of each canola line.

### 2.4. DNA Extraction and Microbiome Profiling

Total genomic DNA was extracted from the soil mixture and rhizosphere soil using a DNeasy PowerSoil Pro Kit (Qiagen Inc., Toronto, ON, Canada), while a DNeasy Plant Pro Kit (Qiagen Inc.) was used for DNA extraction from seed and root samples. Root and seed samples (~100 mg) were disrupted using a mortar and pestle with liquid nitrogen. Rhizosphere soil and soil mixture samples (~250 mg) were homogenized with a lysis solution in Qiagen PowerBead tubes using a Vortex-Genie2 (Scientific Industries, Bohemia, NY, USA) at 2700 rpm for 10 min. DNA extraction was completed following the manufacturer’s protocols. DNA yield was quantified using the Qubit DNA HS Assay Kit (Thermo Fisher Scientific, Waltham, MA, USA). Aliquots of the DNA were visualized by gel electrophoresis in 1% agarose stained with SYBR safe DNA stain (Invitrogen, Carlsbad, CA, USA) and compared with a DNA mass ladder (Invitrogen) using a Bio-Rad Gel Doc XR System (Bio-Rad Laboratories, Mississauga, ON, Canada).

The extracted DNA from rhizosphere and root endophytic communities was submitted to Omega Bioservices (Norcross, GA, USA) for sequencing of the 16S rRNA (bacterial) and ITS (fungal) regions using the Illumina MiSeq platform (2 × 250 bp paired-ends). The barcode primers used for amplification of the 16S rRNA region were Bakt_341F (CCTACGGGNGGCWGCAG) and Bakt_805R (GACTACHVGGGTATCTAATCC) [27], while the primers used for amplification of the ITS region were ITS1F (CTTGGTCATTTAGAGGAAGTAA) and ITS4 (TCCTCCGCTTATTGATATGC) [28].

### 2.5. Bioinformatics and Statistical Analyses

Microbiome data analyses were performed using QIIME2 v. 2022.8 [29], with minor adjustments based on official tutorials [30]. Sequences from bacterial and fungal communities were processed separately for each plant compartment (soil mixture, rhizosphere soil, root, and seed). Raw paired-end reads were first trimmed with Cutadapt to remove Illumina adaptors, then de-noised using DADA2 (Divisive Amplicon Denoising Algorithm). The QIIME2 taxa filter-table plugin was used to exclude features identified as mitochondria and chloroplasts from bacterial profiles. Data on the number of quality-filtered reads are provided in Appendix A.

Taxonomic assignments for each Amplicon Sequence Variant (ASV) were made using pre-trained Naive Bayes classifiers, with bacterial sequences classified using SILVA v.138 and fungal sequences using UNITE v. 8_99_04.02.2020. Phylogenetic analysis was performed using the QIIME2 align-to-tree-mafft-fasttree plugin to generate a multiple sequence alignment, and a maximum-likelihood tree was inferred with Fasttree. Venn diagrams representing the relative abundance of genera in the soil, rhizosphere, roots, and seeds were created in PowerPoint using data from the abundance tables sorted at the genus level.

Alpha diversity of the bacterial and fungal microbiomes in the rhizosphere and roots was assessed using the Shannon diversity index. Abundance tables generated in QIIME2 were imported into the estimate_richness function of the Phyloseq package [31] in R Studio (v. 2024.04.0). A linear mixed model was applied using the lme4 R package Lme4 v. 1.1-37 [32] to adjust for the blocking effect associated with the use of different combinations of canola lines across the three experimental rounds. The model included fixed effects for clubroot susceptibility, *P. brassicae* inoculation treatment, and time, with season included as a random effect. Statistical significance was determined using Type III sums of squares with Satterthwaite’s method in the lmerTest R package [33], followed by Tukey’s post hoc tests using the multcomp R package [34].

Beta diversity, based on Bray–Curtis dissimilarity metrics, was analyzed using the q2-diversity plugin in QIIME2. Canonical Analysis of Principal Coordinates (CAP) was employed to adjust for season effects on the dissimilarity matrices. For this analysis, the Bray–Curtis distance matrices generated in QIIME2 were used as input for the capscale function in the Vegan package [35] in R. The capscale function model included inoculation treatment and canola lines as explanatory variables, with the blocking effect, derived from the cultivation of different canola line combinations across three experimental rounds, specified as a conditional factor. Ordination plots were generated for each microbial group (bacteria or fungi), plant compartment (rhizosphere or root), and sampling time-point (7, 21, and 35 dai) using the ggplot function in R. The statistical significance of clustering patterns was evaluated using Permutational multivariate analysis of variance (Permanova).

Differential abundance analysis of observed ASVs across treatments was conducted using the Analysis of Composition of Microbiomes (ANCOM), a compositional method for identifying taxa with significantly different relative abundances among treatments [36]. The biased-corrected ANCOM (ANCOM-BC) was used to account for the blocking effect associated with the cultivation of different canola line combinations across the three experimental rounds.

Prior to analysis, low-abundance ASVs (fewer than 5 reads and present in less than 10% of samples) were removed using the filter-features plugin in QIIME2 to improve the accuracy of identifying differentially abundant taxa. The ANCOM-BC was conducted in R, using abundance tables generated in QIIME2 as input. Results were reported as log-fold changes for each significantly different species, comparing the following treatments: Resistant Canola-Control (RC), Resistant Canola-Inoculated (RI), Susceptible Canola-Control (SC), and Susceptible Canola-Inoculated (SI). Analyses included a Bonferroni correction to control the Type I error rate (i.e., the probability of false positives) when performing multiple hypothesis tests. The sequence data are available in the National Center for Biotechnology Information (NCBI) under accession number PRJNA1044771.

### 2.6. Disease Assessment

Three independent biological replicates (24 plants each) were used for clubroot disease rating for each line in all experiments [37], with the canola cultivar ‘Westar’ included as a positive control. Clubroot severity on each plant was assessed at 42 dai using a 0–3 scale [38]. The individual severity scores were then used to calculate a disease severity index (DSI) according to the formula:DSI (%) = [(n1 × 1 + n2 × 2 + n3 × 3)/(n × 3)] × 100
where n1, n2, and n3 represent the number of plants in each severity class, and n is the total number of plants evaluated [1].

## 3. Results

### 3.1. Clubroot Development

The disease assessments conducted at 42 dai indicated a significant impact of *P. brassicae* on CS canola lines, with DSI values ranging from 88% to 99% (Figure 1). In contrast, inoculated CR canola lines exhibited substantially lower DSI values, ranging from 6% to 27%, confirming their resistance to the pathogen. No symptoms were observed in the non-inoculated control plants (DSI = 0%), demonstrating both the effectiveness of the control treatments and the specificity of the inoculation procedure.

### 3.2. Relative Abundance of Fungal and Bacterial Genera

Taxonomic classification of the sequences revealed 203 bacterial and 27 fungal ASVs in the seeds, corresponding to 28 bacterial and 15 fungal genera, respectively. In the soil mixture, 3360 bacterial and 345 fungal ASVs were detected, representing 290 bacterial and 141 fungal genera. The rhizosphere microbiome was the most diverse, containing 36,926 bacterial and 2471 fungal ASVs, which were assigned to 733 bacterial and 364 fungal genera. In the roots, 8034 bacterial and 434 fungal ASVs were identified, corresponding to 480 bacterial and 136 fungal genera.

Venn diagram analysis (Figure 2) indicated that 53 bacterial genera and 12 fungal genera were shared across the soil, rhizosphere, roots, and seeds. A greater number of bacterial and fungal genera were found exclusively among the soil, rhizosphere, and roots, compared to only two genera shared among the soil, rhizosphere, and seeds. No genera were exclusively shared between the soil, roots, and seeds. Additionally, 19 bacterial and 3 fungal genera were shared exclusively among the rhizosphere, roots, and seeds.

The most abundant bacterial genera detected in the seeds were *Pseudomonas* (37%), *Bacillus* (17%), *Rhizobium* (9%), and *Pedobacter* (7%) (Figure 3). *Pedobacter* and *Sporocytophaga* were more prevalent in the susceptible lines, whereas *Bacillus* and *Bryobacter* dominated in the resistant lines. Among fungal taxa in the seeds, the most prevalent genera were *Fusarium* (19%), *Alternaria* (7%), and *Mortierella* (5%). *Fusarium* was more abundant in the susceptible lines, while *Alternaria* was more common in the resistant lines.

In the soil mixture, the predominant bacterial genera were *Bacillus* (8%), *Udaeobacter* (8%), *Sphingomonas* (6%), and *Streptomyces* (5%), while the dominant fungal genera were *Pseudeurotium* (35%), *Mortierella* (13%), *Phialemonium* (12%), and *Solicoccozyma* (6%) (Figure 4).

The taxonomic data for microbial genera associated with the rhizosphere and roots was pooled by calculating the average relative abundance (%) of these genera for each factor (i.e., canola line, inoculation treatment, and sampling time) to visualize bacterial and fungal community profiles across these variables (Appendix A). The rhizosphere soil associated with canola lines displayed similar bacterial genera profiles (Appendix A), with the most abundant genera being *Sphingomonas* (6%), *Flavobacterium* (5%), *Udaeobacter* (5%), *Pseudarthrobacter* (4%), and *Bacillus* (4%). In contrast, the bacterial genera distribution within the roots varied among canola lines, sampling times, and *P. brassicae* inoculation treatments (Appendix A). The most prevalent bacterial genera in the root endosphere included *Pseudomonas* (23%), *Rhizobium* (8%), *Massilia* (6%), *Cellvibrio* (5%), and *Flavobacterium* (5%). Similarly, fungal profiles in both the rhizosphere soil and roots showed significant variation in genera distribution across canola lines, sampling times, and inoculation treatments (Appendix A). The most abundant fungal genera in the rhizosphere were *Mortierella* (50%), *Phialemonium* (9%), *Olpidium* (7%), *Humicola* (5%), and *Fusarium* (5%). In the roots, the predominant fungal genera were *Olpidium* (53%), *Rhizophlyctis* (15%), *Fusarium* (11%), and *Mortierella* (8%).

### 3.3. Diversity of Microbial Communities

The beta diversity of the bacteria and fungi associated with seeds was analyzed using a PCoA ordination of the Bray–Curtis dissimilarity (Figure 5). Bacterial profiles showed contrasting differences between CS and CR canola lines along axis1 (Figure 4), suggesting that clubroot susceptibility may influence bacterial diversity. However, PERMANOVA results indicated that neither canola line nor clubroot susceptibility had a significant impact on bacterial beta diversity, although *p*-values for clubroot susceptibility were below 0.1, suggesting a potential effect (Table 1). In contrast, analysis of fungal community structure revealed distinct clustering: CR lines TR254 and TR277 grouped together, while CS lines TS245 and TS247 formed a separate cluster (Figure 5). Additionally, fungal profiles of canola line TR271 differed markedly from those of the other lines. PERMANOVA confirmed that both canola line and clubroot susceptibility significantly influenced fungal community composition (Table 1).

Analysis of bacterial alpha diversity in the rhizosphere microbiome showed that Shannon index values were influenced only by the clubroot susceptibility of the canola lines; neither *P. brassicae* inoculation nor time had a significant effect. Bacterial diversity was higher in the rhizosphere of CS lines compared to CR lines. In contrast, root-associated bacterial diversity exhibited an interaction between clubroot susceptibility and time (Figure 6, Table 2), with Shannon diversity declining in the roots of CR lines at 35 dai (Figure 6).

For the fungal microbiome, Shannon index analysis revealed a significant interaction between clubroot susceptibility and time in the rhizosphere communities (Figure 7, Table 2). CS lines showed lower fungal diversity compared to CR lines, especially at 21 dai (Figure 6). Both clubroot susceptibility and *P. brassicae* inoculation affected root fungal diversity (Figure 7, Table 2). Shannon diversity values were higher in controls and inoculated CR lines compared to their respective CS counterparts. However, no significant differences were observed between control and inoculated roots within either CS or CR groups (Figure 7).

### 3.4. Effect of P. brassicae Inoculation on Bacterial and Fungal Communities

The ANCOM was conducted to assess the response of bacterial and fungal communities to *P. brassicae* inoculation in CR and CS canola lines. Individual comparison graphs were generated for each microbial group (bacteria or fungi), plant compartment (rhizosphere or root), and sampling time (7, 21, and 35 dai).

In this study, we hypothesized that species showing higher abundance in resistant canola lines following *P. brassicae* inoculation, compared to inoculated susceptible lines, may play a potential role in clubroot control. Therefore, ANCOM was used to compare microbial abundance across four treatments: Resistant Canola-Control (RC), Resistant Canola-Inoculated (RI), Susceptible Canola-Control (SC), and Susceptible Canola-Inoculated (SI). Two separate analyses were performed with either RC or SC as the reference group for comparisons.

Results are reported as log-fold changes for species that exhibited significant differences between any compared treatments. Overall findings are summarized in Table 3, Table 4, Table 5 and Table 6, highlighting bacterial and fungal species that were less abundant in the SI group compared to the RI group, as well as taxa with equal or lower abundance in the inoculated groups (RI, SI) relative to their respective controls (RC, SC). Some taxa showed no significant differences depending on the reference group used in the analysis.

Differential abundance analysis of bacterial communities in the rhizosphere identified 17 species with negative log-fold changes in their relative abundance in susceptible lines inoculated with *P. brassicae*. These species also showed negative or zero changes compared to uninoculated controls (Table 3). Most bacteria exhibiting negative changes between the Susceptible Inoculated (SI) and Resistant Inoculated (RI) groups were detected at 7 and 21 dai, while only two species showed negative changes at 35 dai. None of the bacterial species in the rhizosphere exhibited significant changes at multiple sampling times.

In the roots, 18 species showed negative log-fold changes in susceptible inoculated lines, with changes also being negative or zero compared to uninoculated controls (Table 4). Unlike the rhizosphere, most root-associated bacteria exhibiting negative changes between SI and RI were detected at 21 and 35 dai, with only three species showing negative changes at 7 dai. Notably, *Acidovorax facilis* was significant at both 21 and 35 dai, *Cellvibrio fibrivorans* at 7 and 21 dai, and *Cupriavidus agavae* at 7 and 35 dai.

For fungal communities in the rhizosphere, the ANCOM-BC identified 10 species that showed negative log-fold changes in susceptible canola lines inoculated with *P. brassicae* (Table 5). Most of these fungi exhibited reduced abundance in the SI vs. RI comparison at 21 and 35 dai, while only two species showed negative changes at 7 dai. Notably, *Aspergillus luteorubrus* and *Paecilomyces penicilliformis* were significant at 7 and 21 dai.

In the roots, six fungal species showed negative log-fold changes in the susceptible inoculated lines compared to resistant lines (Table 6). However, none of these root-associated fungal species exhibited significant changes across multiple sampling times.

## 4. Discussion

Clubroot remains a persistent threat to canola production, and the development of resistant cultivars has proven to be an essential and effective strategy for protecting canola against *P. brassicae* infection [5,39]. This study provided insights into the prevalence and distribution of specific bacterial and fungal groups within the soil, rhizosphere, root, and seed microbiomes in canola lines with differing susceptibility to clubroot. Analysis of microbial abundance revealed that several bacterial and fungal genera were exclusively shared among the rhizosphere, roots, and seeds (Figure 2). Bacterial genera such as *Pseudomonas* and *Rhizobium* were prevalent in both seeds and roots, while *Bacillus* was a dominant genus in the seeds and rhizosphere (Figure 3 and Appendix A). These findings are consistent with previous reports identifying *Bacillus*, *Pseudomonas*, and members of the Rhizobiaceae as common constituents of the canola seed microbiome [40,41]. Moreover, strains of *Pseudomonas fluorescens* and *Bacillus cereus* have been shown to be effective seed-treatment agents against damping-off caused by *Pythium* spp. in crops such as canola, safflower, dry pea, and sugar beet [42]. Similarly, seed treatments with *Bacillus subtilis*, in combination with other *Bacillus* isolates, have been successfully employed to manage *Sclerotinia sclerotiorum* in *B. napus* [43].

The beta diversity analysis of the seed microbiome indicated that bacterial and fungal community profiles differed between CS and CR canola lines, suggesting that clubroot susceptibility may influence seed-associated microbial composition (Figure 5). Although the effect of clubroot susceptibility on bacterial beta diversity in seeds was not statistically significant, the *p*-values (*p* < 0.1) indicate a potential trend worth further investigation (Table 1). The clustering of fungal communities by canola susceptibility in the seed microbiome (Figure 4) aligns with previous findings linking the relative abundance of certain fungal genera to disease incidence. For instance, a study of the soil microbiome associated with *Lisianthus* flowers found that genera such as *Aspergillus* and *Mortierella* were only present in fields with low Fusarium wilt severity [44]. Similarly, research on bacterial and fungal community structures in soils cultivated with tobacco in China identified *Sphingomonas* as a dominant genus in disease-suppressive soils [45].

In our study, the differential abundance of key microbial genera in the seeds of CS and CR canola lines suggests that specific seed-associated communities may contribute to a plant’s ability to tolerate *P. brassicae* infection. The higher relative abundance of *Bacillus* in CR lines (Figure 2) may indicate a protective role, as *Bacillus* species are well-known for promoting plant growth [46] and suppressing clubroot [12,47,48]. In contrast, the greater prevalence of *Fusarium* in CS lines (Figure 3) raises the possibility that this genus may exacerbate clubroot symptoms when present alongside *P. brassicae*.

The alpha diversity patterns of microbial communities in the rhizosphere and roots of CS and CR canola lines revealed contrasting responses between bacterial and fungal microbiomes (Figure 6 and Figure 7). For bacterial communities, the Shannon diversity in the rhizosphere was higher in CR lines than in CS lines. However, in the roots, CR lines exhibited a decline in bacterial diversity at 35 dai (Figure 6, Table 2). In contrast, fungal communities associated with CS lines showed lower diversity than those of CR lines in both the rhizosphere and roots (Figure 7, Table 2).

The observed decline in bacterial diversity in CR roots over time contrasts with the typically stable or higher diversity seen in resistant lines, as was observed for fungal communities [49,50]. This suggests that while CR lines may initially support more diverse and stable microbial communities, prolonged disease pressure from *P. brassicae* may still disrupt microbial equilibrium. Differences in microbial diversity between CS and CR lines may also be explained by the variation in root exudation patterns. Resistant and susceptible lines are known to release different root exudates, which serve as carbon sources that influence microbial recruitment, colonization, and community structure [19,51,52].

The impact of *P. brassicae* inoculation on alpha diversity was only evident in the fungal communities associated with roots. In contrast, bacterial alpha diversity in both the rhizosphere and roots, as well as fungal alpha diversity in the rhizosphere, remained unaffected by inoculation (Figure 6 and Figure 7, Table 2). The contrasting patterns in Shannon diversity between CS and CR lines and between bacterial and fungal communities, suggest that the influence of clubroot susceptibility on microbial diversity is both microbe-type and tissue-specific. The greater stability of fungal diversity in CR lines over time may indicate a more resilient fungal community under disease pressure. Meanwhile, the elevated bacterial diversity observed in CS lines could be driven by genotypic differences unrelated to disease response, possibly reflecting traits linked to plant growth promotion or stress adaptation.

The ANCOM revealed differences in bacterial and fungal abundances in the rhizosphere and roots of canola plants in response to *P. brassicae* inoculation, as well as differences between CR and CS lines (Table 3, Table 4, Table 5 and Table 6). Negative log-fold change values in susceptible lines indicate that certain microorganisms were less abundant in the presence of the pathogen, potentially reflecting their role in disease suppression. These findings support our hypothesis that resistant lines may recruit or sustain beneficial microbial communities that help mitigate clubroot severity. Previous research comparing the microbiomes of symptomatic and asymptomatic *B. napus* plants similarly found that many bacteria were exclusive to asymptomatic samples and may provide plant health benefits [53].

Several bacterial genera that were predominant in resistant canola in our study, such as *Acidovorax*, *Bacillus*, *Cellvibrio*, *Chryseobacterium*, *Cupriavidus*, *Cytophaga*, *Duganella*, *Flavobacterium*, *Fluviicola*, *Luteimonas*, *Methylotenera*, *Pedobacter*, *Peredibacter*, and *Ureibacillus*, have previously been detected in the rhizosphere and roots of canola [54,55,56,57,58,59,60]. Similarly, fungal genera such as *Aspergillus*, *Candida*, *Fusicolla*, *Paecilomyces*, and *Rhizophlyctis* have been previously identified in canola-associated environments [61,62,63]. Some of the bacterial taxa that were less abundant in inoculated CS canola lines (Table 3 and Table 4) have previously been identified as potential biocontrol agents in canola and other crops. For example, members of the genus *Acidovorax* have been reported to promote plant growth in barley seedlings by producing secondary metabolites and phytohormones, as well as by competing with plant pathogens [64]. *Bacillus* species, including *B. amyloliquefaciens*, *B. subtilis*, and *B. velezensis*, have been shown to suppress *P. brassicae* infection and reduce clubroot severity in greenhouse trials [12,47,48]. These strains produce hydrolytic enzymes and a range of secondary metabolites with antagonistic and plant growth-promoting properties [65,66].

Other bacterial genera with reported biocontrol or plant-beneficial traits include *Cupriavidus*, which demonstrated significant production of indole-3-acetic acid (IAA), phosphate solubilization, and ACC deaminase activity, enhancing morphological and biochemical traits in *B. napus* seedlings [67]. *Flavobacterium* spp. were found to stimulate root elongation in *Brassica juncea*, even under cadmium stress, through the production of indoles, siderophores, and ACC deaminase [68]. In rice, seed endophytic *Flavobacterium* strains exhibited IAA production, phosphate solubilization, and high tolerance to salinity and osmotic stress [69]. *Rhodococcus* strains have shown the ability to alleviate plant stress via ACC deaminase activity, degrade environmental contaminants, and stimulate lateral root emergence in *Arabidopsis thaliana* [70]. Additionally, the genera *Ureibacillus* and *Peredibacter* include species isolated from compost that have been associated with disease suppression and plant growth promotion, respectively [71].

Several fungal genera that were less abundant in the presence of *P. brassicae* (Table 5 and Table 6) have previously been reported to promote growth and/or suppress disease in canola and other crops. For instance, *Aspergillus* and *Paecilomyces* have demonstrated antagonistic activity against *S. sclerotiorum* in canola, suggesting their potential as biocontrol agents [72,73]. Similarly, *Candida subhashii* has been described as a competitive and antagonistic soil yeast with broad-spectrum activity against multiple plant-pathogenic fungi, supporting its use in biocontrol strategies [74]. In addition, studies have shown that *Linnemannia* species can stimulate aerial growth in *A. thaliana* and influence key hormonal pathways involving auxin, ethylene, and reactive oxygen species [75]. These findings highlight the potential of beneficial fungi not only to enhance plant health and development but also to serve as biological control agents against clubroot and other soilborne diseases.

The shifts in bacterial and fungal community composition over time (Table 3, Table 4, Table 5 and Table 6) highlight the dynamic nature of plant-microbe interactions and suggest that effective disease control strategies should consider these temporal changes. Microbial species that showed negative log-fold changes in susceptible lines at the early (7 dai) and mid (21 dai) stages may provide immediate benefits to the plant. For instance, bacterial taxa such as *Bacillus*, *Cellvibrio*, and *Fluviicola*, along with fungal taxa such as *Aspergillus* and *Paecilomyces*, showed significant changes soon after inoculation, suggesting their role in initial plant defense responses, nutrient cycling, or antagonism toward *P. brassicae*.

In contrast, microbial species that remain active or become significant at later stages (35 dai), such as *Cupriavidus*, *Flavobacterium*, and *Rhodococcus* among bacteria, and *Candida* among fungi, may offer prolonged protection or support under sustained pathogen pressure. Species such as *Acidovorax facilis*, *Aspergillus novofumigatus*, *Cellvibrio fibrivorans*, *Cupriavidus agavae*, and *Paecilomyces penicilliformis*, which were consistently less abundant in susceptible lines across multiple time-points, could exert influence across several phases of infection. These findings reinforce the importance of understanding microbial succession and dynamics in shaping plant responses to clubroot stress and developing targeted microbial-based management strategies.

This study provides an in-depth analysis of how microbial communities in susceptible and resistant canola lines respond to *P. brassicae* inoculation, identifying potential microbial candidates for clubroot biocontrol. By examining bacterial and fungal taxa that differ between resistant and susceptible lines and tracking their temporal dynamics, this research provides insights to support the development of effective bioinoculants for clubroot management. Moreover, integrating plant microbiome profiling with breeding strategies holds promise for guiding the assembly of predictable and functional microbial communities that support plant health and development [76]. Although many of the microbial species detected in this study are already known for their plant growth-promoting or biocontrol properties, our findings emphasize the importance of deploying microbial consortia tailored to the dynamic shifts in root-associated communities following *P. brassicae* infection. The proposed combinations of bacteria and fungi provide a foundation for future research and practical application in integrated clubroot management strategies.

## Figures and Tables

**Figure 1 pathogens-14-00904-f001:**
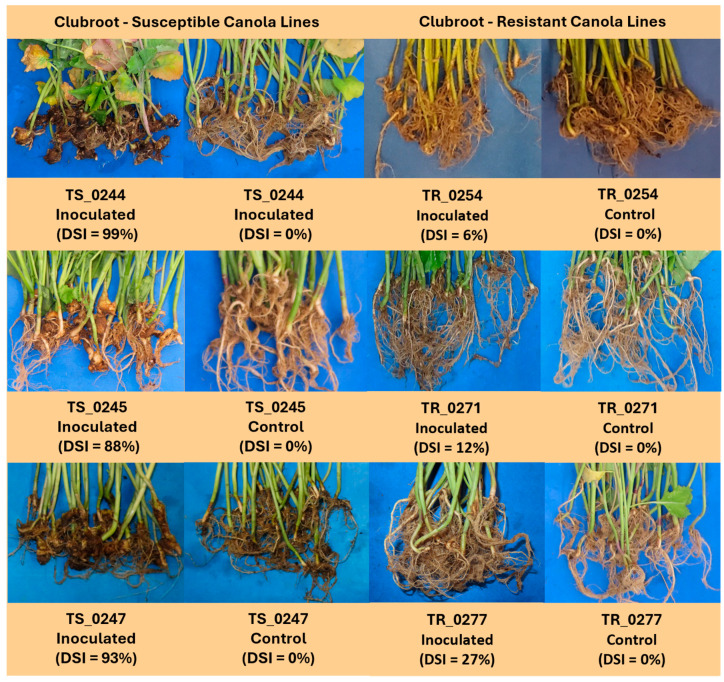
Disease index of clubroot-susceptible (TS244, TS245, and TS247) and clubroot-resistant (TR254, TR271, and TR277) canola lines inoculated with *P. brassicae* (pathotype 3A) harvested at 42 days after inoculation (dai).

**Figure 2 pathogens-14-00904-f002:**
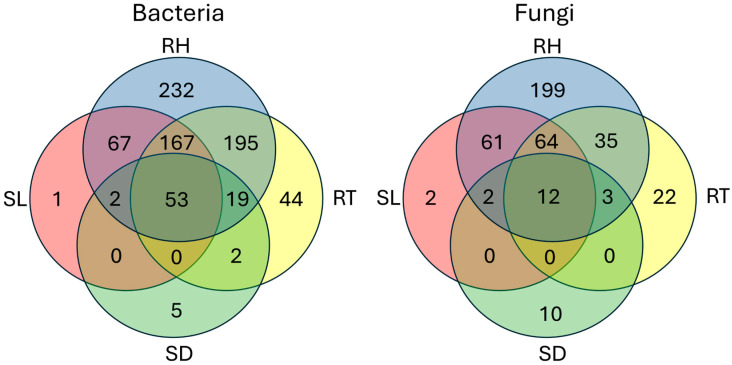
Venn diagram representing bacterial and fungal genera associated with the soil mixture (SL), rhizosphere (RH), roots (RT), and seeds (SD) of three clubroot-susceptible canola lines (TS244, TS245, and TS247), and three clubroot-resistant canola lines (TR254, TR271, and TR277).

**Figure 3 pathogens-14-00904-f003:**
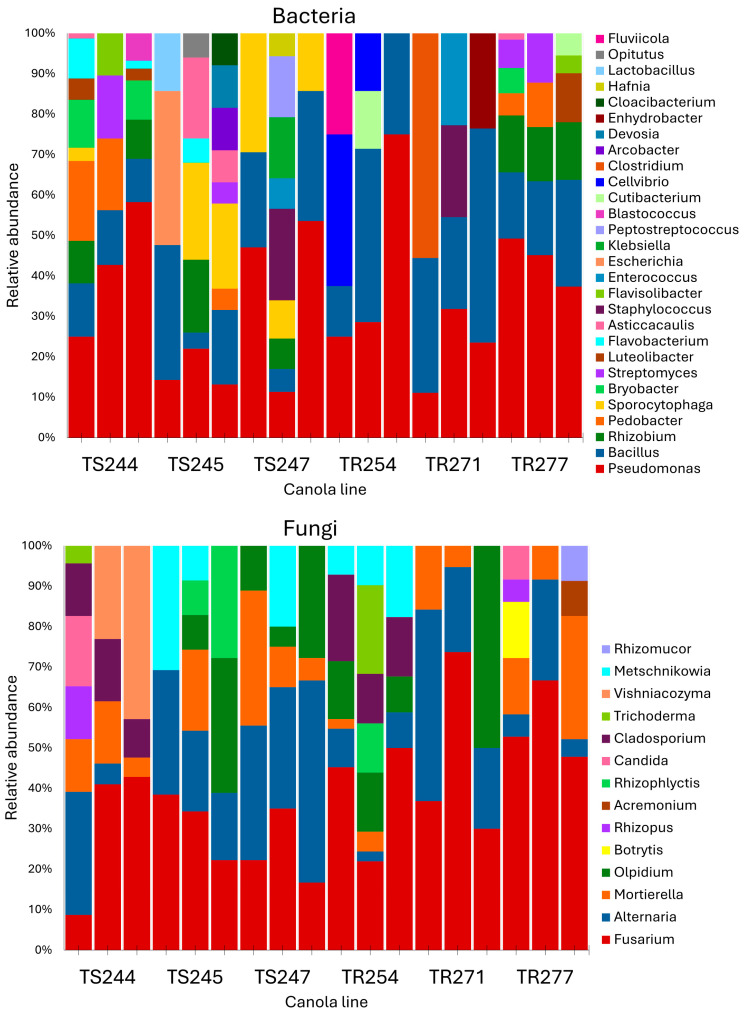
Relative abundance of bacterial and fungal genera in seed samples (N = 3) from clubroot-susceptible canola lines (TS244, TS245, and TS247) and clubroot-resistant canola lines (TR254, TR271, and TR277), as determined by high-throughput sequencing of 16S rRNA and ITS amplicons. Seed samples (1 g) from each canola line were collected before the experiment started to evaluate their baseline microbial composition.

**Figure 4 pathogens-14-00904-f004:**
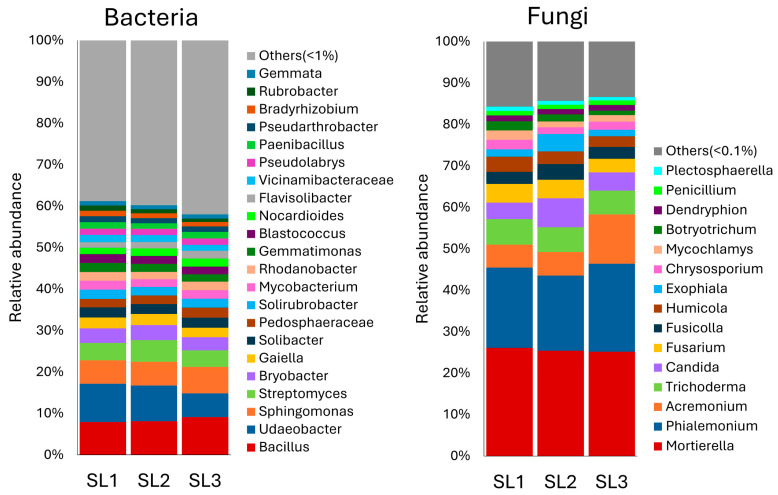
Relative abundance of bacterial and fungal genera in soil mixture samples (SL1, SL2, and SL3) as determined by throughput sequencing of 16S rRNA and ITS amplicons. Soil mixture samples (5 g) were collected before the transplanting of seedlings to evaluate their baseline microbial composition.

**Figure 5 pathogens-14-00904-f005:**
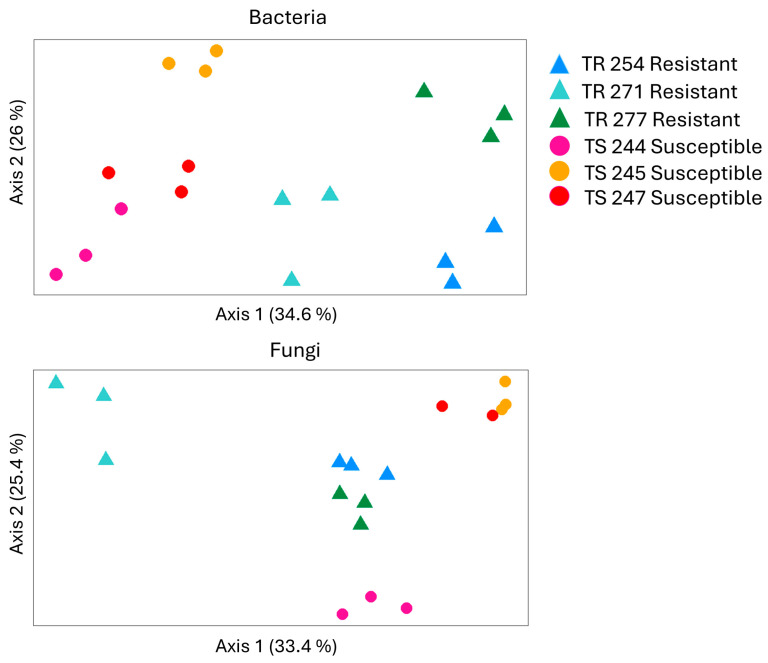
Principal coordinates analysis (Bray–Curtis) of bacterial and fungal communities in seed samples (N = 3) of clubroot-resistant (TR254, TR271, and TR277) and clubroot-susceptible (TS244, TS245, and TS247) canola lines. Seed samples (1 g) from each canola line were collected before the experiment started to evaluate their baseline microbial composition using high-throughput sequencing of 16S rRNA and ITS amplicons.

**Figure 6 pathogens-14-00904-f006:**
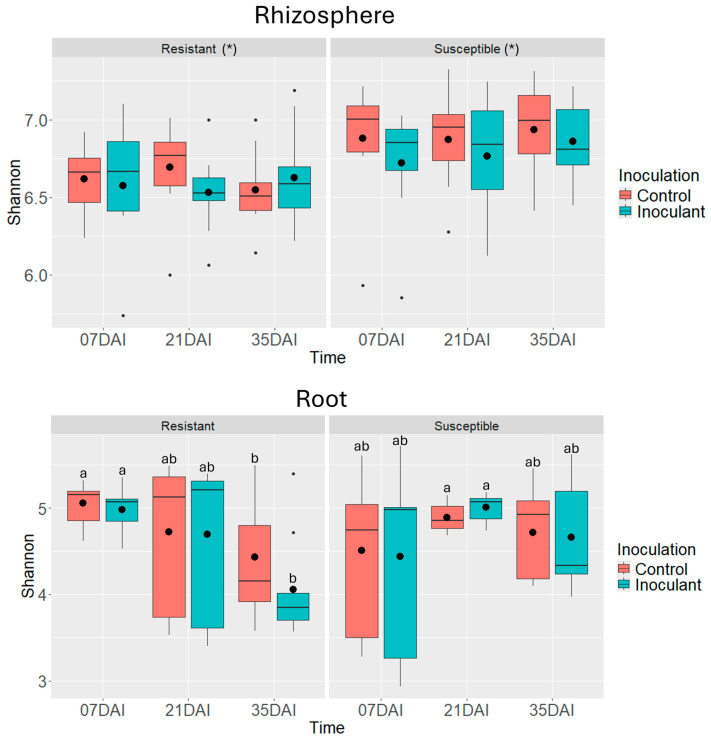
Estimated Shannon’s diversity index of bacterial communities associated with rhizosphere samples and root samples (N = 3) of clubroot-resistant (TR254, TR271, and TR277) and clubroot-susceptible (TS244, TS245, and TS247) canola lines. Plants were harvested at 7, 21, and 35 days after inoculation (dai) with *Plasmodiophora brassicae* (pathotype 3A). Non-inoculated plants served as controls. Different letters denote significant differences (*p* < 0.05) between treatment means based on Tukey’s test. Asterisks (*) indicate statistically significant differences between the overall Shannon index means of the clubroot-susceptible and clubroot-resistant canola lines in the rhizosphere.

**Figure 7 pathogens-14-00904-f007:**
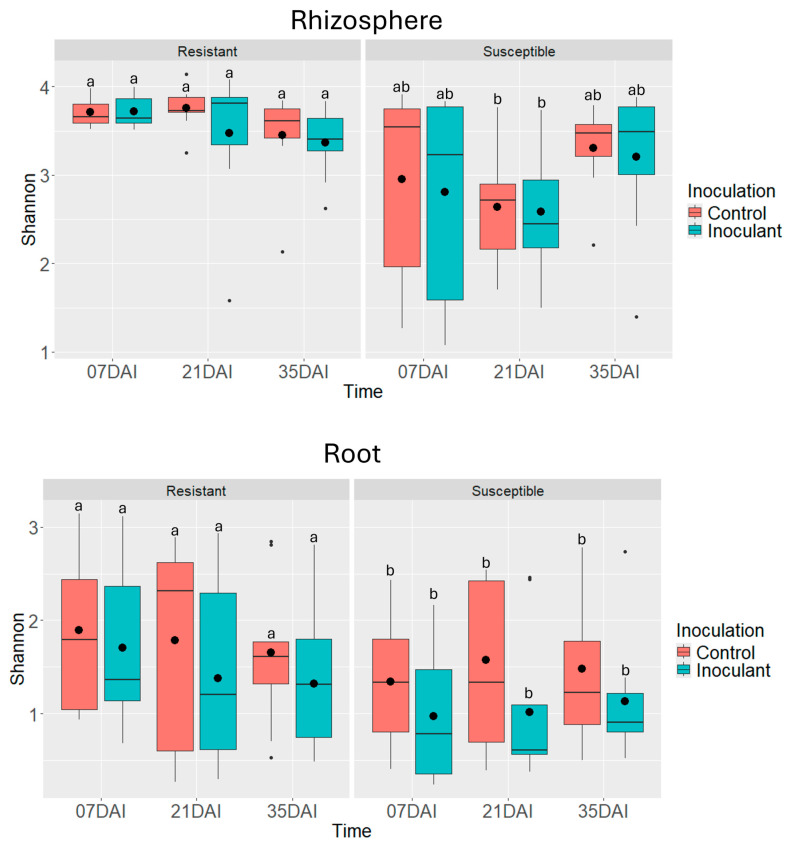
Estimated Shannon’s diversity index of fungal communities associated with rhizosphere samples and root samples (N = 3) of clubroot-resistant (TR254, TR271, and TR277) and clubroot-susceptible (CS) canola lines. Plants were harvested at 7, 21, and 35 days after inoculation (dai) with *Plasmodiophora brassicae* (pathotype 3A). Non-inoculated plants served as controls. Different letters denote significant differences (*p* < 0.05) between treatment means based on Tukey’s test.

**Table 1 pathogens-14-00904-t001:** PERMANOVA analysis (*p*-values) of bacterial and fungal community profiles in seed samples of clubroot-susceptible and clubroot-resistant canola lines.

Factors	Bacteria	Fungi
Canola line	0.78	0.001
Clubroot susceptibility	0.087	0.001

**Table 2 pathogens-14-00904-t002:** Analysis of variance (*p*-values) for the Shannon diversity index of bacterial and fungal community profiles in clubroot-susceptible and clubroot-resistant canola lines.

	Bacteria		Fungi	
Factors	Rhizosphere	Root	Rhizosphere	Root
Clubroot susceptibility	0.0001	0.7048	0.0001	0.0174
*P. brassicae* inoculation	0.1822	0.5149	0.3930	0.0184
Time	0.8292	0.0479	0.3269	0.9111
Clubroot susceptibility ×*P. brassicae* inoculation	0.5280	0.5343	0.9333	0.7022
Clubroot susceptibility × Time	0.6919	0.0041	0.0182	0.4426
*P. brassicae* inoculation × Time	0.5929	0.6931	0.9443	0.8596
*P. brassicae* inoculation × Clubroot susceptibility × Time	0.7388	0.8689	0.8132	0.9735

**Table 3 pathogens-14-00904-t003:** Log-fold changes in the relative abundance of bacterial species in rhizosphere samples (N = 3) of clubroot-resistant (CR) and clubroot-susceptible (CS) canola lines showing differential abundance in response to *Plasmodiophora brassicae* (pathotype 3A) inoculation. Plants were harvested at 7, 21, and 35 days after inoculation (dai).

	Compared to RC		Compared to SC	
Rhizosphere 7 dai	SI−RI	RI−RC	SI−RI	SI−SC
*Acidibrevibacterium fodinaquatile*	−0.78	0	n.s.	n.s.
*Bacillus badius*	−1.02	0	n.s.	n.s.
*Chryseobacterium limigenitum*	−1.11	0	−1.18	0
*Edaphobacter flagellatus*	−0.9	0	−1.09	0
*Flavobacterium panaciterrae*	−0.97	0	n.s.	n.s.
*Fluviicola taffensis*	−1.14	0	−1.69	0
*Methylotenera mobilis*	−1.52	0	−1.74	0
Rhizosphere 21 dai	SI−RI	RI−RC	SI−RI	SI−SC
*Nordella oligomobilis*	n.s.	n.s.	−1.34	0
*Oligoflexus tunisiensis*	−1.4	0	−1.57	0
*Pedobacter panaciterrae*	n.s.	n.s.	−1.3	0
*Peredibacter starrii*	−1.5	0	−1.47	0
*Risungbinella massiliensis*	−1.12	0	−1.14	0
*Staphylococcus capitis*	−1.17	0	n.s.	n.s.
*Undibacter mobilis*	−1.18	0	−1.36	0
*Ureibacillus thermophilus*	−1.04	0	n.s.	n.s.
Rhizosphere 35 dai	SI−RI	RI−RC	SI−RI	SI−SC
*Rhodococcus globerulus*	−1.51	0	n.s.	n.s.
*Williamsia faeni*	n.s.	n.s.	−1.13	0

Note: Log-fold changes in the relative abundance of significantly different bacterial species were assessed using the Analysis of Composition of Microbiomes (ANCOM). Two separate analyses were conducted using either Resistant Canola-Control (RC) or Susceptible Canola-Control (SC) as the reference group. Reported comparisons include Susceptible Canola-Inoculated vs. Resistant Canola-Inoculated (SI–RI), Resistant Canola-Inoculated vs. Resistant Canola-Control (RI–RC), and Susceptible Canola-Inoculated vs. Susceptible Canola-Control (SI–SC). Species marked ‘n.s.’ indicate no statistically significant change in abundance for the corresponding comparison.

**Table 4 pathogens-14-00904-t004:** Log-fold changes in the relative abundance of bacterial species in root samples (N = 3) of clubroot-resistant (CR) and clubroot-susceptible (CS) canola lines showing differential abundance in response to *Plasmodiophora brassicae* (pathotype 3A) inoculation. Plants were harvested at 7, 21, and 35 days after inoculation (dai).

	Compared to RC		Compared to SC	
Root 7 dai	SI−RI	RI−RC	SI−RI	SI−SC
*Cellvibrio fibrivorans*	−2.24 (*)	−2.16	n.s.	n.s.
*Cupriavidus agavae*	−2.14 (*)	0	−2.22 (*)	−2.47
*Ferruginibacter lapsinanis*	n.s.	n.s.	−1.22	0
Root 21 dai	SI−RI	RI−RC	SI−RI	SI−SC
*Acidovorax facilis*	−1.86 (*)	−1.36	n.s.	n.s.
*Cellvibrio fibrivorans*	n.s.	n.s.	−1.86 (*)	−1.89
*Cellvibrio gandavensis*	n.s.	n.s.	−1.86	−1.84
*Corallococcus macrosporus*	−2.35	0	n.s.	n.s.
*Cytophaga hutchinsonii*	n.s.	n.s.	−2.18	−1.67
*Duganella flavida*	−2.03	−1.87	−2.18	−1.46
*Fluviicola kyonggii*	−1.75	0	n.s.	n.s.
*Longitalea arenae*	n.s.	n.s.	−1.05	0
*Ralstonia solanacearum*	n.s.	n.s.	−1.32	0
*Roseateles saccharophilus*	n.s.	n.s.	−1.4	0
*Uliginosibacterium sediminicola*	−1.8	−1.38	n.s.	n.s.
Root 35 dai	SI−RI	RI−RC	SI−RI	SI−SC
*Acidovorax facilis*	−2.04 (*)	0	−2.14	−1.35
*Crocinitomix algicola*	n.s.	n.s.	−1.68	0
*Cupriavidus agavae*	−1.8 (*)	0	−1.9 (*)	0
*Dechloromonas denitrificans*	−2.44	0	−2.13	0
*Flavobacterium panaciterrae*	−1.62	0	n.s.	n.s.
*Herpetosiphon aurantiacus*	n.s.	n.s.	−1.16	0
*Luteimonas notoginsengisoli*	n.s.	n.s.	−1.07	−1.05

Note: Log-fold changes in the relative abundance of significantly different bacterial species were assessed using the Analysis of Composition of Microbiomes (ANCOM). Two separate analyses were conducted, using either Resistant Canola-Control (RC) or Susceptible Canola-Control (SC) as the reference group. Reported comparisons include Susceptible Canola-Inoculated vs. Resistant Canola-Inoculated (SI–RI), Resistant Canola-Inoculated vs. Resistant Canola-Control (RI–RC), and Susceptible Canola-Inoculated vs. Susceptible Canola-Control (SI–SC). (*) Species exhibiting significant changes at different sampling times. Species marked ‘n.s.’ indicate no statistically significant change in abundance for the corresponding comparison.

**Table 5 pathogens-14-00904-t005:** Log-fold changes in the relative abundance of fungal species in rhizosphere samples (N = 3) of clubroot-resistant (CR) and clubroot-susceptible (CS) canola lines showing differential abundance in response to *Plasmodiophora brassicae* (pathotype 3A) inoculation. Plants were harvested at 7, 21, and 35 days after inoculation (dai).

	Compared to RC		Compared to SC	
Rhizosphere 7 dai	SI−RI	RI−RC	SI−RI	SI−SC
*Aspergillus luteorubrus*	−2.14 (*)	0	−2.25 (*)	0
*Paecilomyces penicilliformis*	−2.11 (*)	0	−2.22	0
Rhizosphere 21 dai	SI−RI	RI−RC	SI−RI	SI−SC
*Aspergillus luteorubrus*	−1.34 (*)	0	n.s.	n.s.
*Paecilomyces penicilliformis*	−1.83 (*)	0	−1.51	0
*Phialemonium obovatum*	n.s.	n.s.	−2.01	0
*Pseudogeomyces hebridensis*	n.s.	n.s.	−2.03	0
*Terfezia pseudoleptoderma*	−1.89	0	n.s.	n.s.
Rhizosphere 35 dai	SI−RI	RI−RC	SI−RI	SI−SC
*Alternaria alstroemeriae*	−2.48	0	−2.52	0
*Arachnomyces peruvianus*	−1.55	0	n.s.	n.s.
*Candida subhashii*	−2.18	0	n.s.	n.s.
*Leohumicola minima*	−1.4	0	n.s.	n.s.
*Oidiodendron eucalypti*	−1.52	0	n.s.	n.s.

Note: Log-fold changes in the relative abundance of significantly different fungal species were assessed using the Analysis of Composition of Microbiomes (ANCOM). Two separate analyses were conducted using either Resistant Canola-Control (RC) or Susceptible Canola-Control (SC) as the reference group. Reported comparisons include Susceptible Canola-Inoculated vs. Resistant Canola-Inoculated (SI–RI), Resistant Canola-Inoculated vs. Resistant Canola-Control (RI–RC), and Susceptible Canola-Inoculated vs. Susceptible Canola-Control (SI–SC). (*) Species exhibiting significant changes at different sampling times. Species marked ‘n.s.’ indicate no statistically significant change in abundance for the corresponding comparison.

**Table 6 pathogens-14-00904-t006:** Log-fold changes in the relative abundance of fungal species in root samples (N = 3) of clubroot-resistant (CR) and clubroot-susceptible (CS) canola lines showing differential abundance in response to *Plasmodiophora brassicae* (pathotype 3A) inoculation. Plants were harvested at 7, 21, and 35 days after inoculation (dai).

	Compared to RC		Compared to SC	
Root 7 dai	SI−RI	RI−RC	SI−RI	SI−SC
*Olpidium brassicae*	−2.78	0	0	0
*Rhizophlyctis rosea*	−3.39	0	−3.4	0
Root 21 dai	SI−RI	RI−RC	SI − RI	SI−SC
*Linnemannia fatshederae*	−1.48	0	n.s.	n.s.
*Linnemannia hyalina*	−2.42	0	−1.79	0
*Mortierella globalpina*	−1.73	0	n.s.	n.s.
Root 35DAI	SI−RI	RI−RC	SI−RI	SI−SC
*Fusicolla aquaeductuum*	−2.14	0	n.s.	n.s.

Note: Log-fold changes in the relative abundance of significantly different fungal species were assessed using the Analysis of Composition of Microbiomes (ANCOM). Two separate analyses were conducted using either Resistant Canola-Control (RC) or Susceptible Canola-Control (SC) as the reference group. Reported comparisons include Susceptible Canola-Inoculated vs. Resistant Canola-Inoculated (SI–RI), Resistant Canola-Inoculated vs. Resistant Canola-Control (RI–RC), and Susceptible Canola-Inoculated vs. Susceptible Canola-Control (SI–SC). Species marked ‘n.s.’ indicate no statistically significant change in abundance for the corresponding comparison.

## Data Availability

The original data presented in the study are openly available in NCBI under accession number PRJNA1044771.

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
