# Peer review of "Impact of Plasmodiophora brassicae on Canola Root and Rhizosphere Microbiomes and Its Implications for Clubroot Biocontrol"

_pathogens, 2025, doi:10.3390/pathogens14090904_

Round 1
Reviewer 1 Report
Comments and Suggestions for Authors
Dear Authors,
The aim of this article was to study the influence of Plasmodiophora brassicae, a causal agent of clubroot on the microbial communities in soil, seeds, roots, and rhizosphere of canola, both clubroot-susceptible and clubroot-resistant lines. I found this research to be really comprehensive but well organized and clearly prepared integrating disease assessment with microbiome profiling. The topic is highly relevant, this study addresses very important problem in canola production and gives insights in potentially efficient biological agents against clubroot.
The experiment is well designed and conducted, with clearly organized methodology, detailed protocols and robust data analysis.
The results, particularly the findings that the clubroot-resistant canola lines tend to maintain diverse and stable fungal communities are highly valuable, and I believe they may contribute to future studies aiming to identify potential biocontrol microorganisms. I found this comprehensive research well organized and clearly prepared.
However, I have some comments and suggestions which I believe would improve the article.
L16 – Please change "P. brassicae " to italic
L99 – Please change "B. napus" to italic
L118 – Please check the spore concentration. Should “107” be instead of “10-7”?
L256 – Please change "P. brassicae " to italic
L295 – Please change “Udeobacter” to “Udaeobacter”
L307 – Please remove comma “i.e.,”
L379 - Please change "P. brassicae " to italic
L424-425 – There is some issue with these line numbering as it overlaps with the table.
L606 - Please change "Arabidopsis thaliana” to italic
L630 - Please change "Candida” to italic
Author Response
Comments1: L16 – Please change "P. brassicae " to italic
Response: Thank you for all your suggestions and comments. We have made the correction, as indicated in line 16.
Comments2: L99 – Please change "B. napus" to italic
Response: We have made the correction, as indicated in line 107.
Comments3: L118 – Please check the spore concentration. Should “107” be instead of “10-7”?
Response: Thank you for pointing this out. We have made the correction, as indicated in line 128.
Comments4: L256 – Please change "P. brassicae " to italic
Response: We have made the correction, as indicated in line 271.
Comments5: L295 – Please change “Udeobacter” to “Udaeobacter”
Response: We have made the correction, as indicated in line 313.
Comments6: L307 – Please remove comma “i.e.,”
Response: A comma is optional after the abbreviations “i.e.” and “e.g.”; omission of the comma is British style, while use of a comma is American and Canadian style (please see https://www.noslangues-ourlanguages.gc.ca/en/writing-tips-plus/eg-ie). We have left the comma for now but will follow the technical editor’s recommendations if and when the manuscript goes to print.
Comments7: L379 - Please change "P. brassicae " to italic
Response: We have made the correction, as indicated in 397.
Comments8: L424-425 – There is some issue with these line numbering as it overlaps with the table.
Response: We have made the correction, as indicated in 441.
Comments9: L606 - Please change "Arabidopsis thaliana” to italic
Response: We have made the correction, as indicated in 608.
Comments10: L630 - Please change "Candida” to italic
Response: We have made the correction, as indicated in 632.
Reviewer 2 Report
Comments and Suggestions for Authors
This study focused on describing the Impact of Plasmodiophora brassicae on canola root and rhizosphere microbiomes, however it lacked the experimental evidence of its implications for clubroot biocontrol. It's better to revised the title in order to cover the content of the manuscript.
Abstract:L16-17, it is hard for us to consider this direction. It's better to describe the aim of this study more reasonable.
Introduction: Lacking the informations need to be clarified.
1 Biocontrol is one of the keywords, it's better to illustrate the common biocontrol agents of clubroot fungus, and the underlying mechanism.
2 Why the virulent pathotypes could overcoming clubroot resistance in canola? Whether it is because the alternation of canola root and rhizosphere microbiomes?
Methods: There were several points need to be clarified.
Did you have the Blank controls for analysis the environmental microbiomes?
Growth condition? In the field test or greenhouse conditions?
Lacking the data analysis and figure preparations.
Results: Personally suggestion: Move the Figure S2 into the main manuscript.
Discussion: Too long. Need focused on its potential applications for clubroot biocontrol.
1. The first Para. is no need.
2. There were lots of results could be placed into results, not belong to discussion, especially listed the parameter of microbiomes.
Minor revising suggestion: All the scientific names should be italiated, please check it through the entire manuscript.
Author Response
Comments 1: This study focused on describing the Impact of Plasmodiophora brassicae on canola root and rhizosphere microbiomes, however it lacked the experimental evidence of its implications for clubroot biocontrol. It's better to revised the title in order to cover the content of the manuscript.
Response: Thank you for your suggestions and comments. In this study we analyzed the rhizosphere and root microbiomes of clubroot-susceptible and -resistant canola lines to identify potential microbial candidates for antagonism against P. brassicae. Although we did not test the effects of microbial inoculants on clubroot in this study, our results provide candidate microbes that are currently being evaluated. Therefore, we believe the title accurately reflects the objective of the research.
Comments 2: Abstract:L16-17, it is hard for us to consider this direction. It's better to describe the aim of this study more reasonable.
Response: We have revised the objective for improved clarity. The modified objective is as follows: “This study investigated the impact of P. brassicae infection on the microbial communities of soil, seeds, roots, and the rhizosphere in susceptible and resistant canola lines, with the aim of analyzing host-pathogen-microbiome interactions and identifying microbial taxa potentially associated with disease resistance.” The changes are reflected in line 17.
Comments 3: Biocontrol is one of the keywords, it's better to illustrate the common biocontrol agents of clubroot fungus, and the underlying mechanism.
Response: The third paragraph of the Introduction (lines 62-75) covers the antagonistic bacterial and fungal species that have been investigated for the control of the protist P. brassicae. The mechanisms of biocontrol are the focus of our ongoing investigations and will be reported in future publications; therefore, we have not included information on this topic in the present paper.
Comments 4: Why the virulent pathotypes could overcoming clubroot resistance in canola? Whether it is because the alternation of canola root and rhizosphere microbiomes?
Response: We added the following text to clarify this point: “Pathotype 3A is highly virulent and capable of overcoming resistance in many canola cultivars grown in Canada. Our findings showed that disease severity increased when a P. brassicae isolate, previously cycled multiple times through clubroot-resistant (CR) canola lines, was used to inoculate a CR line. This reduction in resistance is likely driven by the selection and amplification of more virulent pathogen genotypes. The resulting changes in resistance levels also altered the microbial composition of the rhizosphere and root microbiomes, highlighting the need to complement resistance breeding with additional management strategies to sustain the effectiveness of clubroot resistance in canola” This explanation has been included in the Introduction at lines 80-88.
Comments 5: Did you have the Blank controls for analysis the environmental microbiomes?
Response: In this study, we used controls that were not inoculated with P. brassicae. In addition, we included soil mixture and canola seed samples to assess the initial composition of the microbiomes. Information about the controls is provided in the Materials and Methods section (lines 140 and 175-178).
Comments 6: Growth condition? In the field test or greenhouse conditions?
Response: Thank you for pointing this out. The plants were grown on a greenhouse at 22°C with a 16 h photoperiod under natural light supplemented by artificial lighting. This information has been added on line 137.
Comments 7: Lacking the data analysis and figure preparations.
Response: The data analysis and figure preparation procedures are described in the Materials and Methods section (lines 203–257).
Comments 8: Results: Personally suggestion: Move the Figure S2 into the main manuscript.
Response: Figure S2 has been moved to the main manuscript as suggested and is now presented as Figure 1 (line 277).
Comments 9: Discussion: Too long. Need focused on its potential applications for clubroot biocontrol. The first Para. is no need.
Response: The first paragraph of the Discussion has been removed.
Comments 10: There were lots of results could be placed into results, not belong to discussion, especially listed the parameter of microbiomes.
Response: We considered that discussing the effects of P. brassicae inoculation on diversity patterns and the relative abundance of the most prominent genera in the soil and plant compartments provides valuable insights into host–pathogen–microbiome interactions.
Comments 11: Minor revising suggestion: All the scientific names should be italiated, please check it through the entire manuscript.
Response: We have corrected the scientific names throughout the text
Reviewer 3 Report
Comments and Suggestions for Authors
The manuscript is in good shape, the results are well presented, and the figures are adequate.
- Structural Reorganization (Introduction/Discussion): Authors should present the study hypotheses preceding the Objectives section within the Introduction. Consequently, the initial segment of the Discussion must explicitly state whether these hypotheses were partially or fully corroborated.
- The roots were disinfected with NaClO (1.05% v/v) prior to DNA extraction for endomicrobiome analysis. However, immersion in sodium hypochlorite can eliminate epiphytic microorganisms that interact with the plant, biasing the characterization of the endophytic microbiome. No disinfection efficacy test was mentioned (plating rinse water to confirm the absence of contaminants).
I suggest discuss this issue.
- The seed microbiome was evaluated in samples collected prior to the experiment, but it is unknown whether the seeds were treated with fungicides or other agents, which could drastically alter the initial microbial community.
I think this can be considered in the discussion or reported in the methods.
- What was the criterion for choosing the most suitable regression model? Were the AIC/BIC criteria used?
- Multiple comparisons (SI vs. RI, RI vs. RC) were made at different time points (7, 21, 35 days) without adjustment for Type I error (Bonferroni correction). This increases the risk of false positives, maybe Bonferroni correction would be most suitable.
- Discussion: The authors suggest that CR "tend to maintain more stable and diverse microbial communities" (Abstract), but the bacterial data contradict this statement. The discussion does not explore why bacterial diversity is lower in CR, focusing only on fungi. It´s not clear to me. Maybe the authors could approach better this.
Author Response
Comments 1: Structural Reorganization (Introduction/Discussion): Authors should present the study hypotheses preceding the Objectives section within the Introduction. Consequently, the initial segment of the Discussion must explicitly state whether these hypotheses were partially or fully corroborated.
Response: Thank you for all your suggestions and comments. Our hypothesis is stated in the Introduction at line 88, preceding the objective outlined in lines 91–95. In the Discussion, we explicitly address this hypothesis at line 579 with the sentence: “These findings support our hypothesis that resistant lines may recruit or sustain beneficial microbial communities that help mitigate clubroot severity.” We chose to place this statement within the section of the Discussion that focuses on the differential analysis of microbial taxa in response to P. brassicae infection, as we believe it provides better contextual relevance and flow.
Comments 2: The roots were disinfected with NaClO (1.05% v/v) prior to DNA extraction for endomicrobiome analysis. However, immersion in sodium hypochlorite can eliminate epiphytic microorganisms that interact with the plant, biasing the characterization of the endophytic microbiome. No disinfection efficacy test was mentioned (plating rinse water to confirm the absence of contaminants). I suggest discuss this issue.
Response: As suggested, the procedure to test the efficacy of the disinfection method was included in the Material and Methods section (line 171). The effectiveness of sterilization methods in microbiome assessments has been previously discussed by other authors, and we consider it unnecessary to include this topic in our Discussion.
Comments 3: The seed microbiome was evaluated in samples collected prior to the experiment, but it is unknown whether the seeds were treated with fungicides or other agents, which could drastically alter the initial microbial community. I think this can be considered in the discussion or reported in the methods.
Response: The canola seeds used in our study were not treated with any fungicides or other chemical or biological agents. As suggested, this information has been added to the Materials and Methods section (line 111).
Comments 4: What was the criterion for choosing the most suitable regression model? Were the AIC/BIC criteria used?
Response: The regression model was selected based primarily based on the experimental design and biological rationale rather than formal model selection criteria such as AIC or BIC. Specifically, the fixed effects, including clubroot susceptibility, P. brassicae inoculation treatment, and time, were chosen because they directly reflect the main experimental factors under investigation. The random effect for season was included to account for variation across the three experimental rounds, which involved different combinations of canola lines and environmental conditions.
Comments 5: Multiple comparisons (SI vs. RI, RI vs. RC) were made at different time points (7, 21, 35 days) without adjustment for Type I error (Bonferroni correction). This increases the risk of false positives, maybe Bonferroni correction would be most suitable.
Response: The analyses of microbial taxa, including multiple comparisons between SI vs. RI and RI vs. RC, were performed using ANCOM-BC. These analyses included a Bonferroni correction to control the Type I error rate (i.e., the probability of false positives) when performing multiple hypothesis tests. We have clarified this point in the Materials and Methods section (line 254).
Comments 6: Discussion: The authors suggest that CR "tend to maintain more stable and diverse microbial communities" (Abstract), but the bacterial data contradict this statement. The discussion does not explore why bacterial diversity is lower in CR, focusing only on fungi. It´s not clear to me. Maybe the authors could approach better this.
Response: In the Abstract (line 23), the following statement is included: “Diversity analyses of microbial communities revealed that clubroot-resistant canola lines tended to maintain more stable and diverse fungal communities, with a higher Shannon index than susceptible lines.” Bacterial diversity patterns were not mentioned in the Abstract. In the Discussion (line 556), we addressed bacterial diversity as follows: “The observed decline in bacterial diversity in CR roots over time contrasts with the typically stable or higher diversity seen in resistant lines, as was observed for fungal communities. This suggests that while CR lines may initially support more diverse and stable microbial communities, prolonged disease pressure from P. brassicae may still disrupt microbial equilibrium”
Reviewer 4 Report
Comments and Suggestions for Authors
The Manuscript ID: pathogens-3808537 addresses the impact of Plasmodiophora brassicae on canola root and rhizosphere microbiomes and its implications for clubroot biocontrol. The procedures described and materials utilized in their work properly treat the main question tackled by the research; aiming to evaluate the influence of the fungus on the related settings, particularly in the context of clubroot biocontrol. Analysis of Composition of Microbiomes (ANCOM) revealed differences in bacterial and fungal abundances in the rhizosphere and roots of canola plants in response to P. brassicae inoculation, as well as differences between CR and CS lines (Tables 3-6). Negative log-fold change values in susceptible lines indicated that certain microorganisms were less abundant in the presence of the pathogen, potentially pointing to their role in disease suppression. These findings suggest that resistant lines may recruit or sustain beneficial microbial communities that help mitigate clubroot severity. Bacterial taxa such as Bacillus, Cellvibrio, and Fluviicola, along with fungal taxa such as Aspergillus and Paecilomyces, in susceptible lines showed significant changes soon after fungal inoculation, suggesting their role in initial plant defense responses, nutrient cycling, or antagonism toward P. brassicae. Eventually, the study offers an in-depth analysis of how microbial communities in susceptible and resistant canola lines respond to P. brassicae inoculation, identifying potential microbial candidates for clubroot biocontrol.
The subject is worth publication and the authors did a good job. Yet, further insights might improve the study and specific improvements should be considered:
- The study focused on the favorable impact of resistant canola lines on the microbial communities at brassicae inoculation. On the contrary, other researchers could describe how microbial-mediated modulation of host immune responses facilitated parasitism. For example, they discussed the role of Caenorhabditis elegans-protective microbiota to get an insight into the microbial protection of phytonematode parasites in soil. Therefore, to make a good balance, I suggest that the discussion section focuses also on how microbial-mediated modulation of canola susceptible lines responded in favor of P. brassicae parasitism.
- Scientific names when reported for the first time in the text should be written in full with Authority and systematics; e.g. Plasmodiophora brassicae Woronin (Plasmodiophorida: Plasmodiophoridae). And so on….
- I suggest moving Figure S2 from the supplementary to the MS.
- Many typos, misprints, and mistakes were found in the MS and should be corrected, to name but a few:
- In the abstract: “P. brassicae” italic instead of “P. brassicae”; Likewise “B. napus lines that were either” instead of “B. napus lines that were either” and so on for others.
- In several sentences; it is written "The ANCOM analysis was conducted to assess the response of bacterial and fungal…" This phrase is not correct. The correct is "The ANCOM was conducted to assess the response of bacterial and fungal …”
- Never use initials at the beginning of a sentence, e.g. “ANCOM analysis revealed differences in bacterial and fungal abundances in the rhizosphere and roots of canola plants in response to …”.
- Needless to remind to delete that “the accession number of the sequence data has not yet been obtained at the time of submission and will be provided during review.” as it is enough to be included in the supplementary.
- “and resilience to disease [40] (Ning et al. 2020).” Please, delete (Ning et al. 2020)
Author Response
Comments 1: The study focused on the favorable impact of resistant canola lines on the microbial communities at brassicae inoculation. On the contrary, other researchers could describe how microbial-mediated modulation of host immune responses facilitated parasitism. For example, they discussed the role of Caenorhabditis elegans-protective microbiota to get an insight into the microbial protection of phytonematode parasites in soil. Therefore, to make a good balance, I suggest that the discussion section focuses also on how microbial-mediated modulation of canola susceptible lines responded in favor of P. brassicae parasitism.
Response: Thank you for your suggestions and comments. The objective of our study was to assess the effects of P. brassicae on the rhizosphere and roots of CR and CS canola lines, with the goal of identifying antagonistic microbes that could be used for clubroot biocontrol. The information generated in this research can be applied to investigate the role of microbes and microbially induced plant responses in P. brassicae parasitism. Some members of our research team are currently addressing this topic; therefore, we have not included it in the Discussion.
Comments 2: Scientific names when reported for the first time in the text should be written in full with Authority and systematics; e.g. Plasmodiophora brassicae Woronin (Plasmodiophorida: Plasmodiophoridae). And so on….
Response: We have corrected this error. The change is reflected in line 40.
Comments 1: I suggest moving Figure S2 from the supplementary to the MS.
Response: Thank you for this suggestion, which was also raised by another reviewer. We agree and have moved Figure S2 to the main manuscript, where it is now presented as Figure 1 (line 277).
Comments 3: Many typos, misprints, and mistakes were found in the MS and should be corrected, to name but a few: In the abstract: “P. brassicae” italic instead of “P. brassicae”; Likewise “B. napus lines that were either” instead of “B. napus lines that were either” and so on for others.
Response: We have corrected these typos throughout the text.
Comments 4: In several sentences; it is written "The ANCOM analysis was conducted to assess the response of bacterial and fungal…" This phrase is not correct. The correct is "The ANCOM was conducted to assess the response of bacterial and fungal …”
Response: We have corrected these typos throughout the text.
Comments 5: Never use initials at the beginning of a sentence, e.g. “ANCOM analysis revealed differences in bacterial and fungal abundances in the rhizosphere and roots of canola plants in response to …”.
Response: We have corrected this error throughout the text.
Comments 6: Needless to remind to delete that “the accession number of the sequence data has not yet been obtained at the time of submission and will be provided during review.” as it is enough to be included in the supplementary.
Response: We have obtained the accession number, which is now included in line 256.
Comments 7: “and resilience to disease [40] (Ning et al. 2020).” Please, delete (Ning et al. 2020)
Response: The above-mentioned author name has been removed from the text, leaving only the reference number [40] in line 500.
Round 2
Reviewer 2 Report
Comments and Suggestions for Authors
Personal suggestion: carefully revised Discussion section: Too long. Need focused on its potential applications for clubroot biocontrol.
Author Response
Comments 1: Personal suggestion: carefully revised Discussion section: Too long. Need focused on its potential applications for clubroot biocontrol.
Response: Thank you for your suggestion. In response, we have shortened the Discussion section by removing the last three sentences of the first paragraph and the entire second paragraph (lines 487–503 of the previous version), the fourth paragraph (lines 525–527), and the last sentence of the sixth paragraph (lines 546–548). These changes reduced the Discussion from 192 to 161 lines, and we have updated the citation list accordingly. We note, however, that discussing the effects of P. brassicae inoculation on diversity patterns provides valuable insights into host–pathogen–microbiome interactions and, therefore, remains an essential part of the discussion.
